

# From command-line bioinformatics to bioGUI

Markus Joppich and Ralf Zimmer

Department of Informatics, Ludwig-Maximilians-Universität München, Munich, Germany

## ABSTRACT

Bioinformatics is a highly interdisciplinary field providing (bioinformatics) applications for scientists from many disciplines. Installing and starting applications on the command-line (CL) is inconvenient and/or inefficient for many scientists. Nonetheless, most methods are implemented with a command-line interface only. Providing a graphical user interface (GUI) for bioinformatics applications is one step toward routinely making CL-only applications available to more scientists and, thus, toward a more effective interdisciplinary work. With our *bioGUI* framework we address two main problems of using CL bioinformatics applications: First, many tools work on UNIX-systems only, while many scientists use Microsoft Windows. Second, scientists refrain from using CL tools which, however, could well support them in their research. With *bioGUI* install modules and templates, installing and using CL tools is made possible for most scientists—even on Windows, due to *bioGUI*'s support for Windows Subsystem for Linux. In addition, *bioGUI* templates can easily be created, making the *bioGUI* framework highly rewarding for developers. From the *bioGUI* repository it is possible to download, install and use bioinformatics tools with just a few clicks.

## INTRODUCTION

Many advances in bioinformatics rely on sophisticated applications. Examples are Trinity (*Grabherr et al., 2011*) for de novo assembly in conjunction with Trimmomatic (*Bolger, Lohse & Usadel, 2014*), or the HISAT2, StringTie and Ballgown pipeline for transcript-level expression analysis (*Pertea et al., 2016*). These tools have in common, that, locally installed, only a command-line interface (CLI) is provided, implying a burden for many non-computer affine users (*Morais et al., 2018*). Jellyfish (*Marçais & Kingsford, 2011*), Glimmer (*Delcher et al., 2007*) and HMMer (http://hmmer.org) natively run only in UNIX-environments and require a sophisticated setup on Windows. In addition, the installation of command-line (CL) tools is a challenge for non-computer specialists, for example, due to package dependency resolution. This problem has been addressed by the AlgoRun package (*Hosny et al., 2016*), providing a *Docker*-based repository of tools. Being a web-based service, it is limited to web-applicable data sizes, or local data must be made available to the Docker container in the cloud. While AlgoRun has the advantage of

Corresponding authors
Markus Joppich,
joppich@bio.ifi.lmu.de
Ralf Zimmer, zimmer@ifi.lmu.de

processing data anywhere, it relies on *Docker*. Docker may be run either on a local workstation or in the cloud. On a local workstation it can induce incompatibilities with existing software (using Hyper-V on Windows). A cloud-based service may conflict with data privacy guide lines (*Schadt, 2012*), for example, with respect to a possible de-anonymization of patient samples (*Gymrek et al., 2013*). Using Windows Subsystem for Linux (WSL) is often possible in such a scenario: it is provided as an app from the Microsoft Store.

A frequent argument for not providing a graphical user interface (GUI) is the overhead for developing it and the effort to make it really "user centered." Often GUIs are simply deemed unnecessary by application developers. However, one can be sceptical whether non-computer-affine scientists can efficiently use CLIs in their research. In fact, *Albert (2016)* notes that "Bioinformatics, unfortunately, has quite the number of methods that represent the disconnect of the Ivory Tower." *Pavelin et al. (2012)* note that software is often developed without a focus on usability of interfaces (for end-users). While this does not imply that any GUI is helpful, we argue that without a GUI, the otherwise highly sophisticated CL applications are not very useful for some scientists. Besides, a GUI is often more convenient and helps to avoid using wrong parameters, especially if a software is not yet routinely used in a lab. *Smith (2013)* also states that GUI-driven applications make daily work in biology or medical labs easier. Smith remarks that many end-users have a "penchant for point and click," not being able to effectively use CL tools. Still they should have the ability to access and analyse their own data. Many proprietary software solutions address this demand: they allow GUI-based data management, while also being extensible via plug-ins. *Smith (2015)* points out that one of the biggest advantages of such plugins is to combine the power of peer-reviewed algorithms with a user-friendly GUI. Thus, providing a GUI is an important step toward the applicability of methods by end-users. *Visne et al. (2009)* present a universal GUI for *R* aiming to close the gap between *R* developers and GUI-dependent users with limited *R* scripting skills. Additionally, web-based workflow systems, like Galaxy (*Afgan et al., 2016*) or Yabi (*Hunter et al., 2012*) provide means to easily execute (bioinformatics) applications, but aim at more complex workflows. However, both Galaxy and Yabi are designed to be run and maintained by bioinformaticians for several users and are not meant to run on a single, individual basis, like in small labs. More recently *Morais et al. (2018)* stated that the accessibility of bioinformatics applications is one of the main challenges of contemporary biology, and that one of the main problems for users is the struggle of using CLIs. While a GUI does not make an application user-friendly per se, it helps to make it more accessible by lowering the burden to use it (*Xu et al., 2014*; *Visne et al., 2009*; *Anslan et al., 2017*; *Morais et al., 2018*; *Větrovský, Baldrian & Morais, 2018*).

In recent Microsoft Windows operating systems the WSL feature can be activated. This feature provides a native, non-virtualized Ubuntu environment on Windows, allowing to run most applications that also run on Ubuntu. This solves the problems of running unix-tools on Windows. Remaining problems for scientists aiming to run bioinformatics applications thus might be the installation and usage of CL applications.
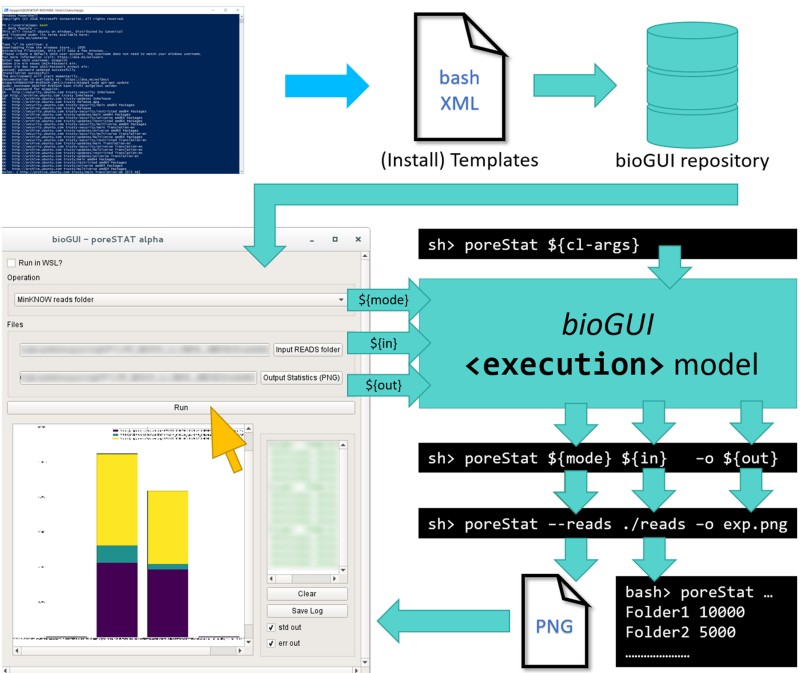

**Figure 1 Only little human interaction is needed to run a CL application from a *bioGUI* template.** An (install) template has to be submitted to the *bioGUI* repository by a developer (blue). The *bioGUI* application (cyan) allows users (yellow) to download templates or install modules and install and use bioinformatics applications. After the user selected/set the input for the (bioinformatics) application using the GUI, the CL arguments to run it are constructed from this input. The application's output (text or images) can be directly displayed in *bioGUI*.   

Here, *bioGUI*, an open-source cross-platform framework for making CL applications more accessible via a GUI, is presented. It uses a XML-based domain-specific language (DSL) for template definition, which lowers the initial effort to create a GUI. *bioGUI* templates for CL applications can easily be scripted. Combined with install modules they provide an efficient and convenient method to deploy bioinformatics applications on Microsoft Windows (via WSL), Mac OS and Linux. *bioGUI* also addresses protocol/parameter management by saving filled out templates, enabling easy reproducibility of data analyses (Fig. 1).

## METHODS

This section first summarizes existing GUI-based systems, then covers the use-case study we performed and goes into detail of how *bioGUI* works.

### Existing (workflow) systems

There are several (workflow) systems already available. Most prominent in bioinformatics are the Galaxy server and Yabi. In addition, workflow specification languages such as the common workflow language (CWL) or Nextflow exist. These workflows do not directly compare to *bioGUI* because they (usually) require a server infrastructure and are not aimed to run on a local computer. However, they have in common that no CLI is needed to run (bioinformatics) applications.

With the R Gui Generator (RGG) a general GUI framework for R already exists. Recently, specialised GUI frameworks, like SEED 2 (*Větrovský, Baldrian & Morais, 2018*) or RNA CoMPASS (*Xu et al., 2014*), have been presented.

### Galaxy and Yabi

The Galaxy server is a well known workflow system in bioinformatics (*Afgan et al., 2016*). While *bioGUI* does not aim to be a workflow system like Galaxy, for example, allowing data management, there are similarities. For instance, Galaxy also provides a (web-based) GUI for its workflows. However, all data to be processed by Galaxy must either be on the server itself or uploaded to a location that is reachable by the server. Galaxy can access cloud storages, but classified data may not be uploaded to such storages as pointed out in the introduction. Additionally, Galaxy requires UNIX knowledge to be installed and does not provide a binary for installation. Galaxy is not cross-platform compatible (Microsoft Windows is supported through WSL but still requires UNIX knowledge). Galaxy users provide Docker containers for Galaxy, where a local storage can be mounted.

Another framework providing similar options is Yabi (*Hunter et al., 2012*). Yabi is only distributed using a Docker container.

### Nextflow and DolphinNext

The combination of Nextflow https://www.nextflow.io/ and DolphinNext https://github.com/UMMS-Biocore/dolphinnext is similar to Galaxy or Yabi. While Nextflow is a DSL for describing general workflows (lacking a GUI definition), DolphinNext provides the web-based user interface (UI) which enables a convenient usage of Nextflow workflows. Nextflow requires a POSIX system architecture and may or may not run on Microsoft Windows using *Cygwin (2019)*. DolphinNext resembles a lot the Galaxy framework, which can make use of CWL workflows, however, focuses on a deployment in a cluster environment. It is unknown whether or not both systems work on WSL.

### Common workflow language

The CWL (*Amstutz et al., 2016*) is a new standard for workflow definition and defines a DSL. In this language, inputs, input-types as well as the corresponding parameters are stored. Additionally, inputs can have a help text included.

Using the bio.tools ToolDog software (*Hillion et al., 2017*), CWL workflows can be generated and exported for many bioinformatics applications. An advantage of using bio.tools is the automatic annotation and description of input and outputs. Unfortunately, for many packages no CWL workflows have been deposited.

### SEED 2 and bioinformatics through windows

In contrast to the previously mentioned tools, SEED 2 (*Větrovský, Baldrian & Morais, 2018*) and Bioinformatics through windows (BTW) (*Morais et al., 2018*) do not focus on running complex workflows in a cluster environment. Instead, these focus on specific tasks which can be run on regular laptop computers. SEED 2 focuses on amplicon high-throughput sequencing data analyses. On the other hand BTW follows the same concept, but focuses on the analysis of marker gene data and does not provide a GUI for

this task. SEED 2 provides a GUI to perform the relevant analyses fast and conveniently, while BTW focuses on the usability of UNIX CL tools on Windows.

### RGG & AutoIt

RGG was developed as a general GUI framework for R applications (*Visne et al., 2009*). It uses XML files to specify the input fields for the graphical representation. When the user has set all options, the GUI is translated into an R script for execution. The execution output can also be retrieved from the RGGRunner application. The RGG software is limited to R scripts, but still the authors expressed their hope that providing GUI for analytical pipelines could "help to bridge the gap between the R developer community and GUI-dependent users" (*Visne et al., 2009*).

In contrast to RGG, *AutoIt (2018)* is a general automation framework which, similar to *bioGUI*, allows the definition of a GUI as well as a task that is executed according to this input. In contrast to AutoIt, *bioGUI* is cross-platform compatible, supports WSL and provides install modules for bioinformatics applications.

### Comparison to bioGUI

*bioGUI* is not a classical workflow system like Galaxy, CWL or DolphinNext with Nextflow. *bioGUI* is not meant to run many tasks nor to run in a cluster environment. Moreover, *bioGUI* does not share the philosophy of having a (compute) cluster setup to run analyses in a repeated fashion. *bioGUI* is meant to enable the user to perform bioinformatics analysis at their work place. With *bioGUI* we aim to provide low effort usage of bioinformatics applications, without the need to setup a complicated environment. Finally this allows to easily compare different methods on collected experimental data.

*bioGUI* finds its niche as a generalisation of the concepts introduced by *Větrovský, Baldrian & Morais (2018)* and *Morais et al. (2018)*. SEED 2 provides a GUI such that a broad public has access to sophisticated and well-known bioinformatics CL applications in the context of amplicon analysis. Similar concepts, yet differently implemented, are provided by RNA CoMPASS (*Xu et al., 2014*) for pathogen-host transcriptome analysis or PipeCraft (*Anslan et al., 2017*). Here, custom (web-)UIs let the user interact with their specialised pipelines. RGG (*Visne et al., 2009*) offers a general GUI framework for R applications only. *bioGUI* offers a similar framework, which is applicable to any (Unix) application. In both, RGG and *bioGUI*, users/developers specify the visual elements in a XML file. This XML file is then interpreted and translated into a GUI within an application (RGGRunner or *bioGUI*, respectively) which also shows the output of the script.

The *bioGUI* framework extends the concepts presented by RGG and SEED 2, for instance, to general applications, and improves accessibility to these applications by providing install modules.

## Use-case study

One of the main goals we had in mind when developing *bioGUI* is to create a powerful framework, which is easy-to-use for scientists/users and which does not create significant overhead for the application developer. In order to study this, we introduce two classes of possible users: The first class represents a general user of the software who generally
prefers a GUI for performing a research task, for example, data analysis after sequencing. The second class describes a software developer releasing an application of a new algorithm to solve the alignment of sequencing reads. This class thus depicts a typical *developer*.

From these two use-cases (see also Appendix section "Use-cases") we identify several requirements/goals for *bioGUI*:

(1) installing new programs must be simple and should not require system administrators

(2) creating a GUI for a program must not take a lot of time

(3) templates must bring a basic GUI to run the programs, output must not be interpreted

(4) templates must be saveable for later re-use and reference, and also searchable

(5) the system must be lightweight (runtime overhead, disk-space) to even run on laptops

(6) installing a program may require additional (protected) external files

Finally, we developed a paper mockup with which we went through the anticipated workflow of the user. We identified several input components and features the *bioGUI* program has to include (Fig. A1).

## bioGUI approach

"The accessibility of bioinformatics applications is crucial and a challenge of contemporary biology" (*Morais et al., 2018*). Particularly the usage of CLIs poses a problem. Since most bioinformatics applications require the execution of commands on the CL for installation (such as for compilation, adding dependencies to the path variable, etc.), we estimate that also the installation poses a problem.

During the use-case study, and interviews with wet-lab scientists without a computational background (Q Emslander, 2019, personal communication; L Jimenez, 2019, personal communication), we found two main problems with bioinformatics applications for scientists which we want to address with *bioGUI*: first the installation of potentially useful applications and second its usage. Both problems have in common, that they are expected to be performed on the CL. A GUI for achieving the respective tasks in bioinformatics (and beyond) is missing.

Especially the first task, installing bioinformatics applications on a user's machine, poses a few problems. Most bioinformatics applications are written for a UNIX operating system, like Linux or Mac OS, while in general Microsoft Windows is the dominant operating system. In order to overcome this problem, *bioGUI* makes use of WSL on Windows. Even if the user's OS is already Unix-like, using the CL to install software might be strugglesome. Thus, in order to support *all* users, *bioGUI* uses a cross-platform approach. *bioGUI* is developed in C++ using the Qt framework.

The general workflow for any program using *bioGUI* is shown in Fig. 1. Given a CL application, the software developer (blue) writes the specific template in a XML-based DSL and can then make this template available, for example, in the *bioGUI repository* (cyan). Such templates can be automatically retrieved by *bioGUI*. Upon selection of a template by the user, *bioGUI* displays the input mask as defined in the template. When the

user (yellow) has filled in all parameters, the parameters are collected by *bioGUI* and assembled into CL arguments which are used to execute the original CL-only application. Upon completion, simple results (like text-output or images) can be shown in *bioGUI* directly, or an external application is opened.

## Install modules

*Install modules* are designed to install applications such that *bioGUI* can access them. Essentially, install modules are bash scripts which allow an automatic installation of applications into a predefined location. For this purpose, install modules receive several arguments from *bioGUI* when launched, for example, where to install the application to, the *sudo* password to fetch packages via a system's package manager (e.g., aptitude, conda, . . . ), whether the application should be made available to the user via the system's PATH variable, etc. Install modules download and install applications and make them available to the user and *bioGUI*. However, some applications cannot be simply downloaded, but are distributed by installers. For this purpose, the install module template can be extended by further input fields. These must be specified by *bioGUI* elements and their values are added to the end of the CL arguments of the install module. An install module can then execute the referenced installer.

Finally, an install module should contain the specification of its *bioGUI* template and could hard-code the path to the installed application. Other constant values, which can already be derived during the installation (e.g., absolute paths to dependencies), could also be defined in the template during this stage.

## *bioGUI* templates

*bioGUI* templates are the actual end-user-interface to programs. A *bioGUI* template defines the look and functions of the UI. Thus it can define how the CL-application is called (with corresponding parameters).

Each *bioGUI* template consists of two parts (Fig. A2). The first part (`<window>` model) defines the visual appearance of the GUI. The second part (`<execution>` model) defines the processing logic of the template. Input values from the GUI components are collected and assembled (e.g., pre-/post-processing steps) to call CL applications. As part of this assembly, input values from the GUI may be transformed using (multiple) predefined nodes. Concatenations are possible using the `<add>` node, and constant values can be inserted using the `<const>` node. System environment properties, such as the operating system, the computer's IP address or specific directories can be collected using the `<env>` node. If the regular nodes are not sufficient, for example, because more complex string manipulations should be made (see use-case study), *script* nodes may also accept functions written in *LUA* (*Lua, 2019*) or *JavaScript* (*JavaScript, 2019*).

In general, the execution part infers a network with inputs (e.g., GUI elements, other nodes within the execution part) and actions (if, add, . . . ). For example, the execution network for an application with many sub-commands is exemplarily shown in Fig. A3.

The time to template varies with the application as well as the number of options to be included. A simple template, like the one for MS-EmpiRe (*Ammar et al., 2019*), can be

created within 10 min. More comprehensive templates, like the one for HISAT2, usually take about 30 min. Time can be saved if only the most important command line options are shown in the GUI. This can be achieved by adding an "optional parameters" input field, where users can insert CL arguments themselves. This is, for instance, shown in the *wtdbg2* (*Ruan & Li, 2019*) and *spades* (*Bankevich et al., 2012*) templates. Adding the install part to a template usually can be done within 15–30 min, depending on how detailed the build process is documented. The creation of an install module thus takes approximately 1 h.

### *bioGUI* integration with CWL and argparse

The CWL (*Amstutz et al., 2016*) only describes the CL workflow and neither provides a GUI nor means to install the desired tool. Due to this more general specification, CWL fits most problems, but specific annotations of inputs, explanations or the embedding of images is not supported in CWL.

While developers can always create templates manually, *bioGUI* supports developers by offering a template generator from CWL templates or python3 argparse CL parsers. Since there are already many CWL templates available for bioinformatics CL applications, CWL files can be used as a base to automatically generate *bioGUI* templates from. Using the *bioGUI template generator* for argparse, it is also possible to automatically generate templates from CWL files (making use of the cwl2argparse program provided by CWL). Our generator takes as input the argparse parser or CWL file and creates input elements for all elements. In case the type of an input is unclear or not supported, the generator falls back to a regular text-input element.

## RESULTS

### bioGUI templates

Currently more than 25 (install) modules exist for *bioGUI*. These represent basically three groups of bioinformatics tasks: next-generation sequencing data analysis and transcriptomics, long read sequencing analysis and assembly as well as more general sequence analysis. In general these install modules will install the respective application on the local machine. The Circlator (*Hunt et al., 2015*) template allows to pull and use the corresponding Docker image. The available tools, as well as their respective categorization, are listed in Table 1.

### Benchmarking bioGUI templates

Our benchmark comprises of four tasks. The first task is to assemble a bacterial genome from Oxford Nanopore long reads, for which the Minimap2 (*Li, 2018*)/miniasm (*Li, 2016*)/ Racon (*Vaser et al., 2017*) pipeline (available as install module from *bioGUI*) is used. The second task is the quantification of reads from a yeast mRNA sequencing project using Oxford Nanpore Reads and Illumina Reads (EMBL ENA studies PRJNA398797 (MinION) and SAMN00849440 (Illumina)). The quantification is performed using featureCounts from the subread package (*Liao, Smyth & Shi, 2014*). The third task uses these results to compute differential gene expression. Differential gene expression analysis is performed

**Table 1 List of available templates and install modules (starting with Install) for *bioGUI*.**

| Module name | Task | Install module | |
| --- | --- | --- | --- |
| | | WSL and Ubuntu | Mac OS |
| First Time Mac OS Setup | Initialisation | – | ✓ |
| First Time Ubuntu/WSL/apt-get Setup | Initialisation | ✓ | – |
| Install Ballgown v1.0.1 (*Pertea et al., 2016*) | NGS transcriptomics | ✓ | |
| Install Bowtie1 (*Langmead et al., 2009*) | NGS | ✓ | |
| Install Bowtie2 v2.2.9 (*Langmead & Salzberg, 2012*) | NGS | ✓ | ✓ |
| Install bwa v0.7.17 (*Li & Durbin, 2009*) | NGS | ✓ | ✓ |
| Install canu (gitHub, *Koren et al. (2017)* | Assembly | ✓ | |
| Install featureCounts (*Liao, Smyth & Shi, 2014*) | NGS transcriptomics | ✓ | ✓ |
| Install glimmer302b (*Delcher et al., 2007*) | Genome annotation | ✓ | |
| Install graphmap (*Sović et al., 2016*) | Long read sequencing | ✓ | ✓ |
| Install albacore (pip wheel, ONT) | Long read sequencing | ✓ | |
| Install guppy (linux tar.gz, ONT) | Long read sequencing | ✓ | |
| Install hisat2 (*Kim et al., 2019*) | NGS transcriptomics | ✓ | ✓ |
| Install hmmer-3.1b2 (*Wheeler & Eddy, 2013*) | Sequence analysis | ✓ | |
| Install jellyfish-2.2.6 (*Marçais & Kingsford, 2011*) | NGS | ✓ | |
| Install minimap2/miniasm/racon (gitHub) | Assembly (long-read) | ✓ | ✓ |
| Install MS-EmpiRe (*Ammar et al., 2019*) | NGS transcriptomics | ✓ | ✓ |
| Install PureSeqTM (*Wang et al., 2019*) | Sequence analysis | ✓ | |
| Install rMATS-3.2.5 (*Shen et al., 2014*) | NGS transcriptomics | ✓ | |
| Install rnahybrid (*Rehmsmeier et al., 2004*) | Sequence analysis | ✓ | ✓ |
| Install RSEM v1.3.0 (*Li & Dewey, 2014*) | NGS transcriptomics | ✓ | |
| Install samtools-1.3.1 (*Li et al., 2009*) | NGS | ✓ | ✓ |
| Install SPAdes v3.13.0 (*Bankevich et al., 2012*) | Assembly (hybrid) | ✓ | ✓ |
| Install StringTie v1.3.0 (*Pertea et al., 2016*) | NGS transcriptomics | ✓ | |
| Install Top Monitor (ssh example) | Technical demo | ✓ | ✓ |
| Install Trimmomatic v0.36 (*Bolger, Lohse & Usadel, 2014*) | NGS | ✓ | |
| Install wtdbg2 (*Ruan & Li, 2019*) | Assembly (long-read) | ✓ | × |
| Template Circlator (*Hunt et al., 2015*) | Assembly | ✓ | ✓ |

**Note:**
Tools marked with ✓ provide an install module for the operating system of the respective column.

using MS-EmpiRe in R (install module available, *Ammar et al. (2019)*). Finally the fourth task uses RNAhybrid to predict miRNA binding sites (1,978 murine miRNAs) in 170 sequences of each 200 nt.

The results are shown in Table 2. The given runtimes are wall clock times. The peak RAM consumption has been sampled from the process viewer on the given operating systems (Task Manager on Windows, top on Linux and Mac OS).

## DISCUSSION

*bioGUI* is a framework for easy GUI-based usage of CL applications in the life sciences. Using *bioGUI*, high-quality CL applications can be made accessible to as many researchers

**Table 2 Benchmarking results for the four selected tasks (see *Benchmarking bioGUI templates* within the Results section) on the described hardware (see Table A1). All runs are started via bioGUI.**

| Task | Linux server | | Lenovo T470 | | Surface book | | MacBook air | |
|---|---|---|---|---|---|---|---|---|
| | Time | Peak RAM | Time | Peak RAM | Time | Peak RAM | Time | Peak RAM |
| Assembly | 10:12 min | 6.8 GB | 23:00 min | 6.5 GB | 30:00 min | 6.5 GB | 44:30 min | 6.5 GB |
| featureCounts (MinION) | 00:38 min | 20 MB | 00:54 min | 18 MB | 01:12 min | 18 MB | 01:30 min | 18 MB |
| featureCounts (Illumina) | 01:13 min | 28 MB | 01:41 min | 25 MB | 02:02 min | 25 MB | 02:30 min | 25 MB |
| DE quantification (MinION) | 00:19 min | 0.7 GB | 00:25 min | 0.6 GB | 00:28 min | 0.6 GB | 00:42 min | 0.6 GB |
| DE quantification (Illumina) | 00:14 min | 0.5 GB | 00:19 min | 0.4 GB | 00:20 min | 0.4 GB | 00:31 min | 0.4 GB |
| RNAhybrid | 07:35 min | 19 MB | 23:00 min | 18 MB | 13:00 min | 18 MB | 16:55 min | 18 MB |

as possible. This is achieved by lowering the hurdles to overcome for using bioinformatics applications, particularly on Windows.

## Use-case analysis

Our use-case analysis (Appendix section "Use-cases") has revealed several requirements for *bioGUI* (see the section "Methods") to enable the user to perform the sequencing analysis and to allow the developer a fast template creation (Fig. 2).

An easy installation (goal 1) is given through the availability of install modules, which can be downloaded from the *bioGUI repository* and started via a GUI. These also allow additional inputs (e.g., Python wheels for albacore, goal 6).

The install modules combine the installation of an application and the creation of the actual GUI template. If the developers employ automatic testing of their software (e.g., build checks with Travis (https://travis-ci.org/)), the install part resembles a Travis container setup (goal 2): dependencies and the application itself are installed into an operating system. Even if not, most bioinformaticians extensively use Ubuntu and/or bash-scripts. Thus writing a script to install dependencies is not a significantly hard workload. We have reached a seamless and time-efficient creation of templates using an XML-based DSL. XML is particularly helpful as it allows to specify hierarchies and attributes to objects. Using our template generator for CWL and python3-argparse, *bioGUI* templates can be created even faster (goal 2). The templates are highly flexible in the creation of CL parameters, also due to providing script nodes. By providing install modules and templates, high-quality open-source bioinformatics applications become more accessible to the community.

The *bioGUI* application is cross-platform compatible and only requires few MBs of disk space (goal 5). *bioGUI* implements several possibilities to execute applications (see Fig. A4). In general, the only runtime overhead involved is the creation of a bash-process which starts the actual program with the assembled CL arguments (goal 5). *bioGUI*, being a local stand-alone application, has the possibility to target both, locally installed and web-based applications, reachable within a controlled environment and with large data. In addition, *bioGUI* also supports the use of Docker containers, for cases where all other options fail.

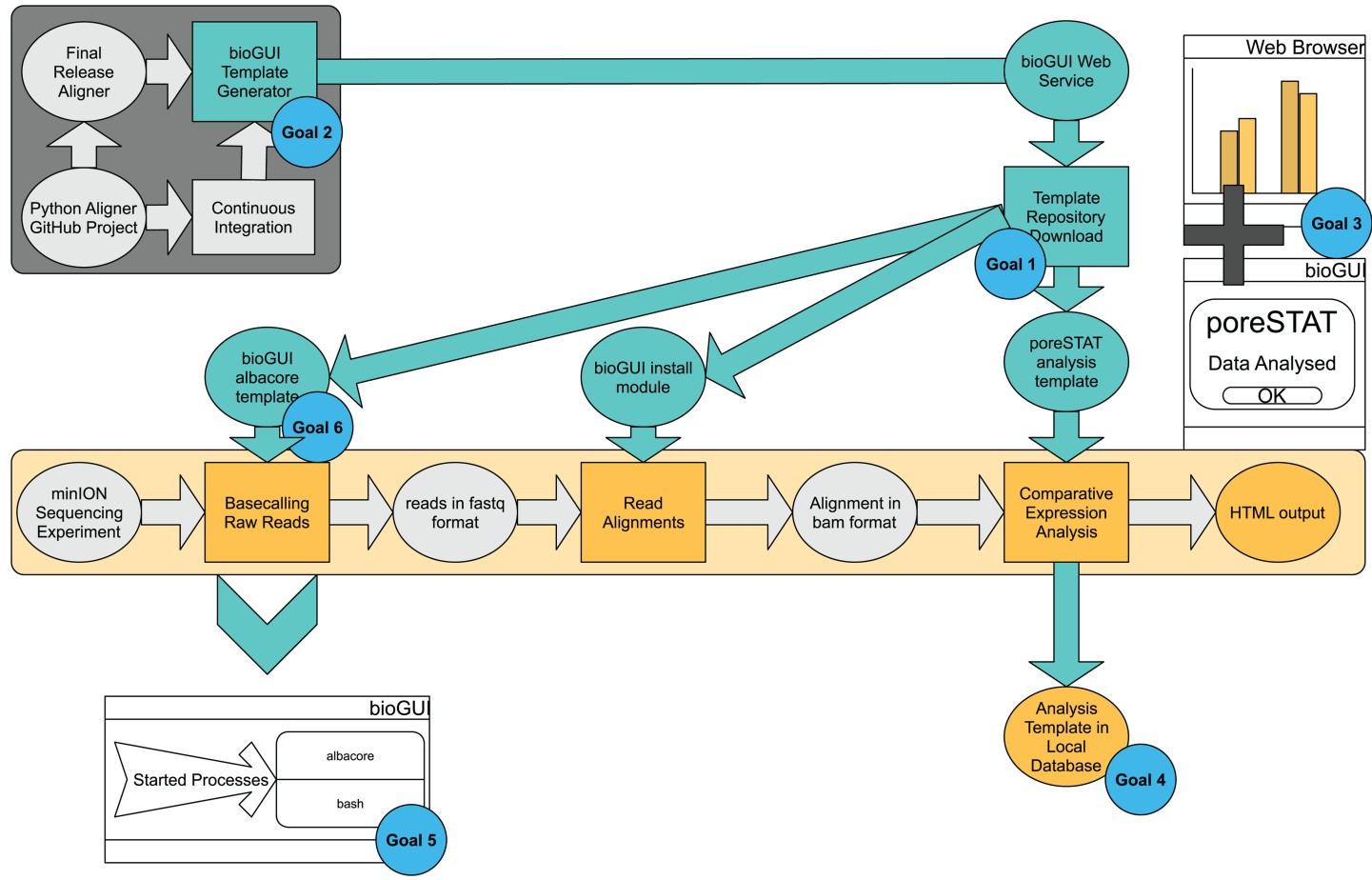

**Figure 2** *bioGUI* **use-case study, from a developer's and user's perspective, performed on an exemplary RNAseq analysis workflow.** The dark-gray underlayed tasks represent the developer's tasks, and the bright-yellow part represents the analysis pipeline the user wants to execute. Tasks requiring user-action are shown as rectangles and intermediate results are shown in ellipses. Cyan ellipses denote solutions/results (e.g., template repository) offered by *bioGUI*. *bioGUI* starts sub-processes for each task, such that the overhead for any started processes is as small as possible. Upon finishing a task or pipeline, *bioGUI* can display a notification and open generated output.

The UI can be made easily understandable (goal 3). Using text-labels, the user can get help on inputs (if specified by the developer), links can be used to provide further information and most importantly tool-tips could also hint the user to which information is needed at a certain step.

Finally, filled out templates can be saved via the *Save Template* button in *bioGUI*, and all available templates can be filtered (goal 4). This enables to keep track of performed analyses, and makes results more reproducible, because parameters are saved. Having the possibility to save templates also allows to easily repeat analysis with the same parameters. Additionally, using the *bioGUI* repository, templates can easily be shared among users, making it easier to standardize runs among different users or even institutes.

An anonymous survey (with 10 participants) about common problems in using bioinformatics software has been conducted among colleagues ($n$ = 4) and under-graduate

students or collaborators from life sciences ($n = 6$, short: collaborators). The results are available in the Appendix section "User Survey."

We asked "What were the most struggle-some tasks in accessing and using the software?" referring to recently used bioinformatics software by the participants. Eight of the 10 participants answered that finding parameters or using the software has been strugglesome. This shows that the selected goals for *bioGUI* address actual problems faced by both experts and regular users. We further asked the participants to install and use *graphmap* (*Sović et al., 2016*), which has been selected because it is reasonably easy to install and use. First the participants were asked to install the tool using the CLI as well as via *bioGUI*. For this task, all instructions have been provided. The question "Has the installation process been easy?" (0 = No, 5 = Yes) has been answered with an average score of 4.4 for the CLI and 4.8 for *bioGUI*. Then the participants were asked to align the given reads against a given reference genome—this time without giving the instructions. Again we asked "Has it been easy to align the reads?" (0 = No, 5 = Yes). Here the CLI scored a 3.5 on average and *bioGUI* a 4.9. (Fig. A6). This coincidences with the answer to our question "Overall: Which interface was easier to use in your opinion?" (0 = CLI, 5 = GUI). Here the average score has been 4. Bioinformaticians and collaborators answered differently: the average bioinformatician has been undecided on which interface has been easier to use (average 3), but non-bioinformaticians preferred the GUI over the CLI (average 4.5).

The survey indicates that there are problems with bioinformatics software regarding installation and usage of CLI tools. These problems can be reduced by providing a GUI for these programs. The more experienced a user is on the CL, the less impact a GUI has. But particularly for non-experts on the CL, a GUI makes it easier to use a program.

## bioGUI repository

We provide a repository of preconfigured templates on our website (Fig. A5), where authors and users can search for and browse existing templates, or submit new ones. *bioGUI* can access uploaded templates and save them directly for use. Specifically for WSL and Ubuntu users install modules are provided, which take care for dependency resolution and install applications (locally) into the user's home. This currently works in any environment using the *aptitude (apt)* package manager, but users can submit templates which also support other environments, since install modules are versatile bash scripts. On Mac OS, some install modules support Homebrew for template installation. Install modules download and potentially severely alter a system (especially if the *sudo* password is supplied). Thus, submitted install modules are manually curated and are only accessible when no security threat has been identified.

The major goal of *bioGUI* is to enable any scientist to use bioinformatics applications. While we extend the repository on a regular basis depending on our own use, users can also request new templates for applications relevant to them.

## Availability and extensibility

*bioGUI* is open-source software and users or institutions can either use our global *bioGUI* repository or deploy a custom repository, for example, one which is only reachable within an institution.

*bioGUI* is available on GitHub (https://github.com/mjoppich/bioGUI). Both source code as well as pre-built binary distributions (for Microsoft Windows, Linux and Mac OS) are available. While *bioGUI* will run on any Linux distribution, install modules currently use mainly aptitude as package manager (e.g., Ubuntu, debian-based distributions). If used on Windows, the same applies for the used WSL-application (Ubuntu 18.04 is recommended). *bioGUI* has been tested on Microsoft Windows 10, build 17763. On Mac OS *bioGUI* uses *Homebrew* (https://brew.sh/) to install dependencies. *Homebrew* does not support a silent, non-interactive installation: the user has to install *Homebrew* before running the *First time setup for Mac OS* install module which will then install the most common dependencies.

While a number of use cases and corresponding components are already included in bioGUI, we encourage users to contribute on GitHub by either pushing their own extensions, or opening feature requests. Further documentation (installation & setup guide, how to write templates) is also available via ReadTheDocs (https://biogui.readthedocs.io/en/latest/index.html).

## Benchmarking bioGUI templates

*bioGUI* starts a sub-process for each executed program. Thus, the only overhead created by *bioGUI* itself is the one for running the GUI, which creates less than 1% CPU usage, allocates less than 50 MB and only performs IO operations when loading a new template (assessed via Sysinternals Process Explorer (*Microsoft Sysinternals, 2019*)).

Nonetheless we have been interested in demonstrating that many bioinformatics tasks do not require a dedicated server setup but can be performed on regular laptop computers. We thus benchmark four typical tasks performed using *bioGUI*.

The selected tasks allow a good overview of different demands: Tasks 1 (assembly) and 3 (differential expression analysis) are CPU-bound tasks, while tasks 2 (feature counting) and 4 (miRNA target prediction) are IO-bound. Particularly task 2 has a high load of read operations, and task 4 has a high demand of write operations. We compare these tasks on a dedicated (Linux) server, one (rather) powerful Lenovo T470p laptop computer, one Surface Book laptop computer (resource-wise a typical laboratory laptop) as well as one MacBook Air. The computer specifications are listed in the Table A1 and results are shown in Table 2. Even though we have not included the alignment of the Illumina yeast reads in this benchmark, it should be noted that this task also runs well on laptop computers. On the Lenovo laptop the alignment of the SRR453566 sample, consisting of 5,725,730 paired reads, has a peak RAM consumption of 34 MB and took 13:50 min, while the Surface Book is even faster at less than 8 min. This presumably can be explained due to different SSD speeds.

The results in Table 2 show that even (computational power-wise) lower-end computers can run bioinformatics tools. More interestingly these results show that the WSL allows the execution of interesting bioinformatic tools. It can be seen that WSL is slow for IO operations, but has a comparable speed for in-memory operations. Particularly tools requiring a lot of IO are considerably faster on the Linux Server (Assembly, RNAhybrid), while the computational expensive tools like MS-EmpiRe (*Ammar et al., 2019*) and featureCounts (*Liao, Smyth & Shi, 2014*) run within similar times.

## CONCLUSION

The *bioGUI* framework makes it easy to develop, provide and use GUIs for CL applications. Particularly for non-computer experts, using CLIs is strugglesome. Providing a GUI and/or install modules increases accessibility to high-quality bioinformatics applications for these users. *bioGUI* creates a cross-platform GUI experience for many open-source bioinformatics applications. In particular, *bioGUI* enables the deployment of academic bioinformatics applications to Microsoft Windows workstations and laptops, but also to Linux or Mac OS.

The separation of the GUI components and the program logic allows the creation of templates in two steps. First, the template developer adds input elements to the window and, second, assembles these inputs according to the needs of the application back into CL arguments. This way almost any CL application can be used with a GUI, enabling many more researchers to use open-source tools. Providing install modules to make UNIX applications available to Microsoft Windows users (via WSL) supports this goal.

*bioGUI* can not always replace dedicated GUIs. A tailored UI will still be more usable and user-friendly than any generic solution can be. We experienced this in our use-case: certain tasks (e.g., selecting options) require special solutions, let alone from displaying or interpreting the results. However, especially with the install module concept we aim to provide a seamless installation and create the possibility to run CL applications by all scientists. Using the *bioGUI* framework, simple GUIs can be constructed. But these simple GUIs already help to make bioinformatic tools more accessible by making execution and usage of these tools more comfortable.

Using *bioGUI*, it becomes a simple exercise to use supported CL applications from a GUI. Currently, there are already installed modules and templates for more than 25 applications in our *bioGUI* repository. *bioGUI* lowers the burden to use excellent applications, allowing more scientists a better analysis of their data. With *bioGUI* it is not necessary to understand how to use and navigate on the CL; instead, the focus is set on the applications, its method and parameters, and finally the data.

# APPENDIX

## Use-cases

### *Non-computer expert*

Many researchers work in small labs without any significant IT support. The computers in their labs mostly run Microsoft Windows and PhD students often have to bring their own devices (because the institute does not provide such working devices). Particularly in the life sciences, users can profit a lot from existing, open-source software. However, installing major bioinformatics applications on such lab computers often poses a problem: administrators (if existent) have little time to deploy new applications, or there is no support in installing new applications at all. If the users are not computer experts, installing and using command line tools may be strugglesome for them (see Appendix section "User Survey"; *Morais et al., 2018*). While there are users that can use the CL efficiently, the cited literature and our personal experience shows that there are many users who do not feel comfortable on the CL. This does not mean that they don't want to learn it or are incapable of learning it, but their focus simply does not lie in learning to use the CL. Instead, they want to get the results for their data, fast, reliably and troubleless. One of these users is Luisa.

Oxford Nanopore Sequencing is becoming more and more popular, and even the sequencing hardware can be found in more and more biological laboratories, like in Luisa's. Particularly important for MinION-sequencing is the post-processing of the actual raw read data. While in previous versions, base-calling has been directly performed in the cloud by Oxford Nanopore, this has now been pushed back to the client side. Thus, despite having the sequencing data on her laptop, Luisa must still retrieve the sequences herself, using, for instance, the Albacore basecaller (if they don't want to rely on LiveBasecalling). Unfortunately, like many bioinformatics packages, the basecaller only comes as a python CL program. Additionally, the download is only available as a python-wheel, which means there is no UI-based setup available. Luisa thus needs assistance for the installation of the python-wheel as well as starting the basecalling process. After the reads are basecalled, reads need to be aligned to a reference genome. While there exist reference genomes in correct format on her lab computer (or can easily be downloaded from the web), the CL program to map the reads is available only from GitHub to be installed from source. Luisa has troubles using the CL to clone the repository, compile and use the CL application.

Luisa does not require a custom analysis of her data, but wants to initially screen her data in a simple, basic and robust analysis. She is mostly busy in her lab, hence an analysis has to be prepared fast, and parameters should be stored for later reference. For this a local searchable database of saved templates is needed.

### *Software developers*

A developer finished his sequence alignment program. The project is already published on GitHub and in a journal, but only few people start using it. From the issues and feature requests on GitHub it can be seen that mainly other bioinformaticians use the program.

Thus the developer decides that the program should be accessible to more researchers and looks for ways to make the program useable by everyone. Since the program is written in python, it is cross-platform compatible per-se. However, it is noticed that domain-experts do not install and use the program. Thus the developer must look for an easy way to distribute the application and make it accessible to more researchers. The developer's time is limited, having other projects waiting. There is also little support for developing GUI from colleagues, as they have different views on the extent of autonomy a wet-lab researcher should have regarding sequencing analysis.

## bioGUI paper mockup

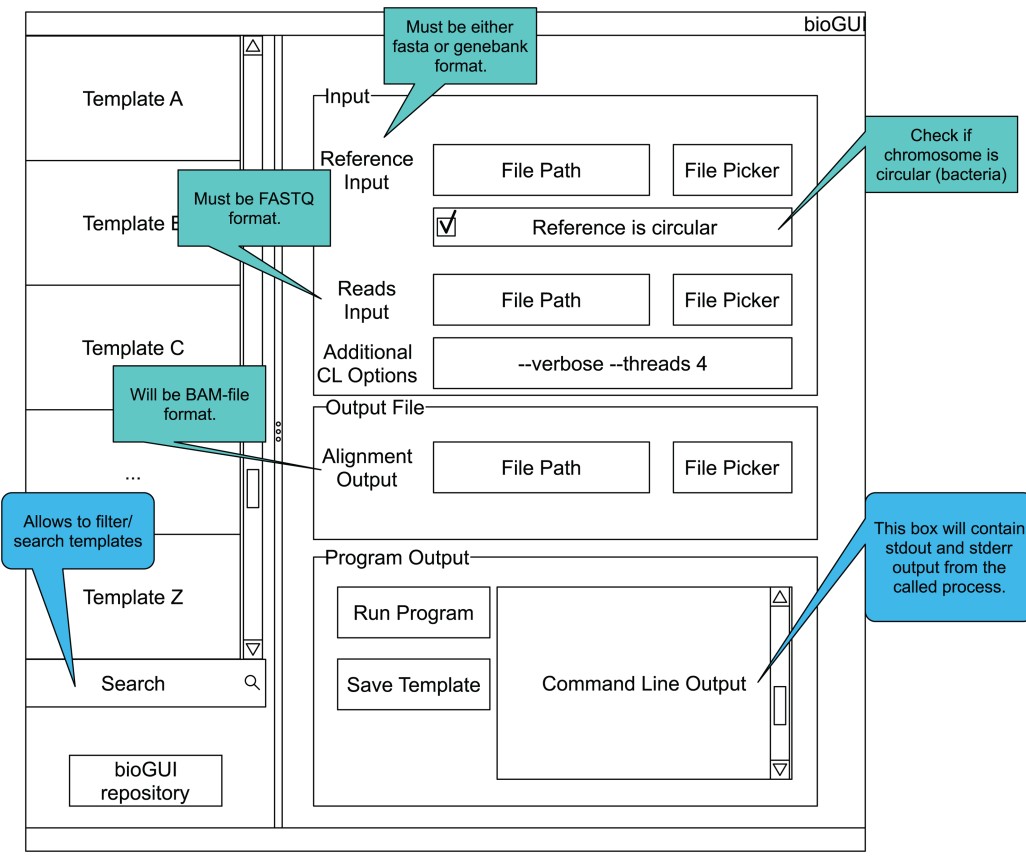

**Figure A1** *bioGUI* **mockup showing the elements a template could be made of.** The GUI has a searchable list of installed templates as well as a link to our repository of templates. The right side is reserved to the currently displayed GUI template. Here a structured view of the available parameters, as well as hints for filling these, is shown to the user. Finally, the user has the possibility to run the program by clicking a button and to see the program's output.

## Extending templates with script nodes

Often it is required to perform string-manipulations (e.g., remove file extensions) for CL arguments. For instance, the example below takes as input a HISAT2 index file, and removes the file extension, such that the index will be accepted by HISAT2. For evaluation of this node, the *evaluate*-function is called with the *argv*-references as

input parameters. The last return-value of the script's call stack is taken as output value of the script -node.

**Listing 1.** bioGUI script node with LUA function example. Upon evaluating this node, the evaluate function will be called with the arguments listed in the argv attribute of the script node.

```
<script id="hisat_index_rel" argv="${hisat_index_rel_raw}">
<![CDATA[
function evaluate(arg1)
        if (string.match(arg1, ".%d.ht2$")) then
            return(string.sub(arg1, 0, arg1:find(".%d.ht2$")-1))
        end
    return(arg1)
    end
]]>
</script>
```

## Evaluating a bioGUI template

In Fig. A2 the process of assembling a CL call from the shown bioGUI template is explained. First, the creation of the `<window>` model (dark gray) will be explained, followed by the creation of the CL arguments using the `<execution>` model (shaded).

The window component consists of four different components, which are grouped in a vertical layout (default for window component). A label describing the input file dialog is placed on the main window, followed by the actual file dialog with ID `input`. Then a group box with title and a checkable status is created, which contains an output file dialog. Finally, the action button, which starts the CL call assembly and the subprocess of this program, and the text output elements are created.

When the user has entered all desired data, and clicks the action button, the execution phase defined by the execution model will be launched. Therefore the program defined in the execute element is started. For this, the parameters (param) must be assembled. Any text within `${var}` is interpreted as a reference to a variable var or the value of a GUI element with id var. Thus, the CL is successively assembled. At first the `${input}` element is interpreted and retrieves the value from the input file dialog as this element matches the id. Next the `${output}` is interpreted. The `${output}` refers to an if construct in the execution part, which compares the value of the element with id os to the string *TRUE* (which is whether the groupbox is checked). If this value is true, this node evaluates to `netcat 192.168.1.100 55025`, otherwise to `tee -a {output file path}`. Finally, the program *sh* is executed with the created CL arguments. For instance, if the group box is checked, the `sh -c "cat inputFile | netcat 192.168.1.100 55025"` will be executed. A full reference of all input types as well as all execution nodes is available online.

In fact, the evaluation of the execution network resembles the simulation of a petri net (Fig. A3). Each node in the execution network is a place, and its modification/function

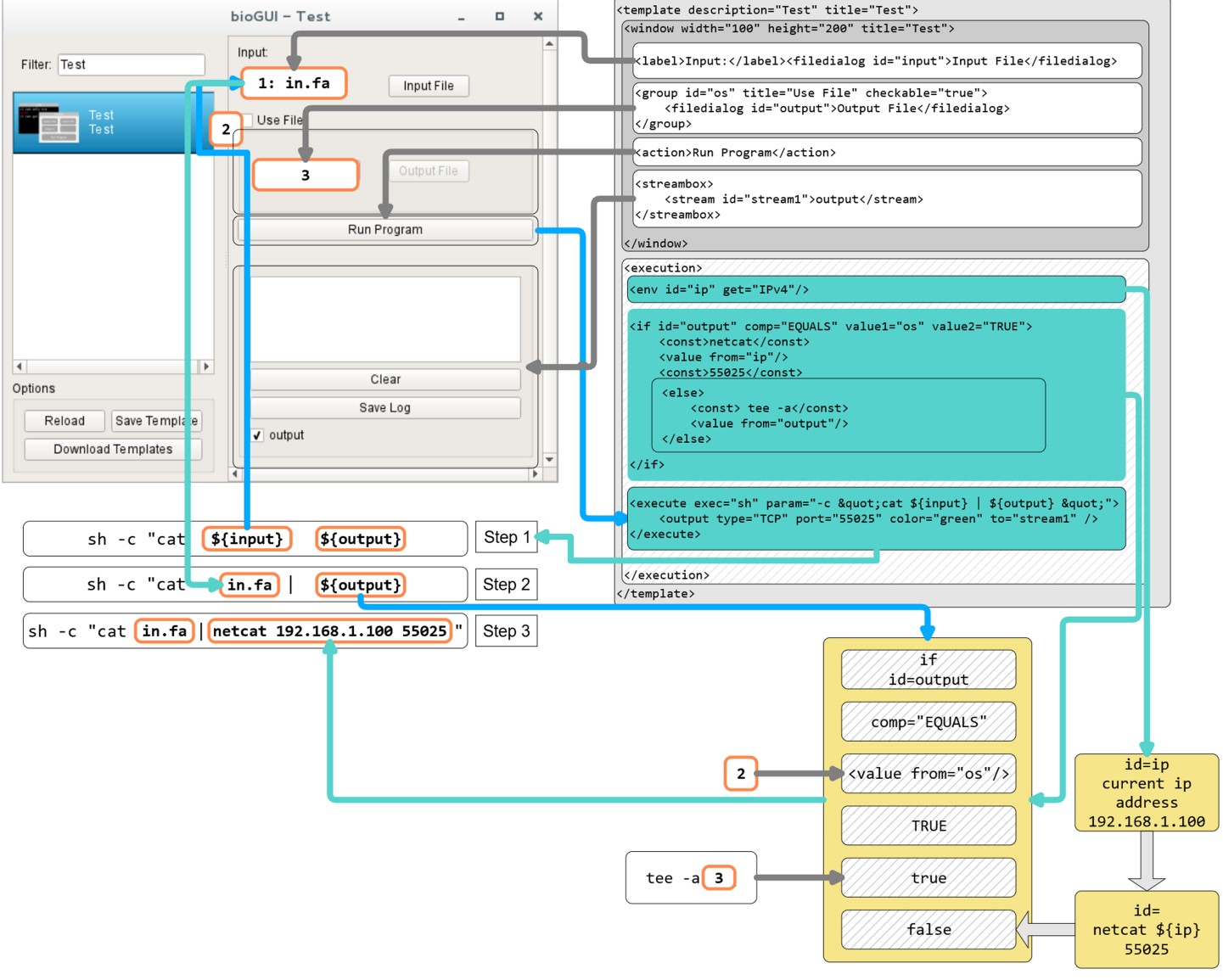

**Figure A2 Template construction and evaluation in *bioGUI*.** First, the dark gray window part is evaluated to create the GUI. Once the user clicks the run button, the execution part of the template (shaded) is executed by constructing and starting the assembled system call. This system call is constructed in three steps by replacing variables with evaluated terms from the user's input. Blue lines indicate the visual element a returned value (cyan lines) is taken from. Helper/intermediate nodes to be evaluated are shown in light yellow.  

is the transition, which requires values for all its input places, to generate the output token.

## Running programs via bioGUI

Program execution via *bioGUI* can be accomplished via different paths, which are shown in Fig. A4. The easiest way is to execute a native program (one that runs natively on the operating system, e.g., Docker). Then all output can be piped to bioGUI to display this to the user. If the host is a Microsoft Windows 10 OS, *bioGUI* can also run Unix programs via WSL. Then the Unix program runs natively in a WSL bash. The resulting output

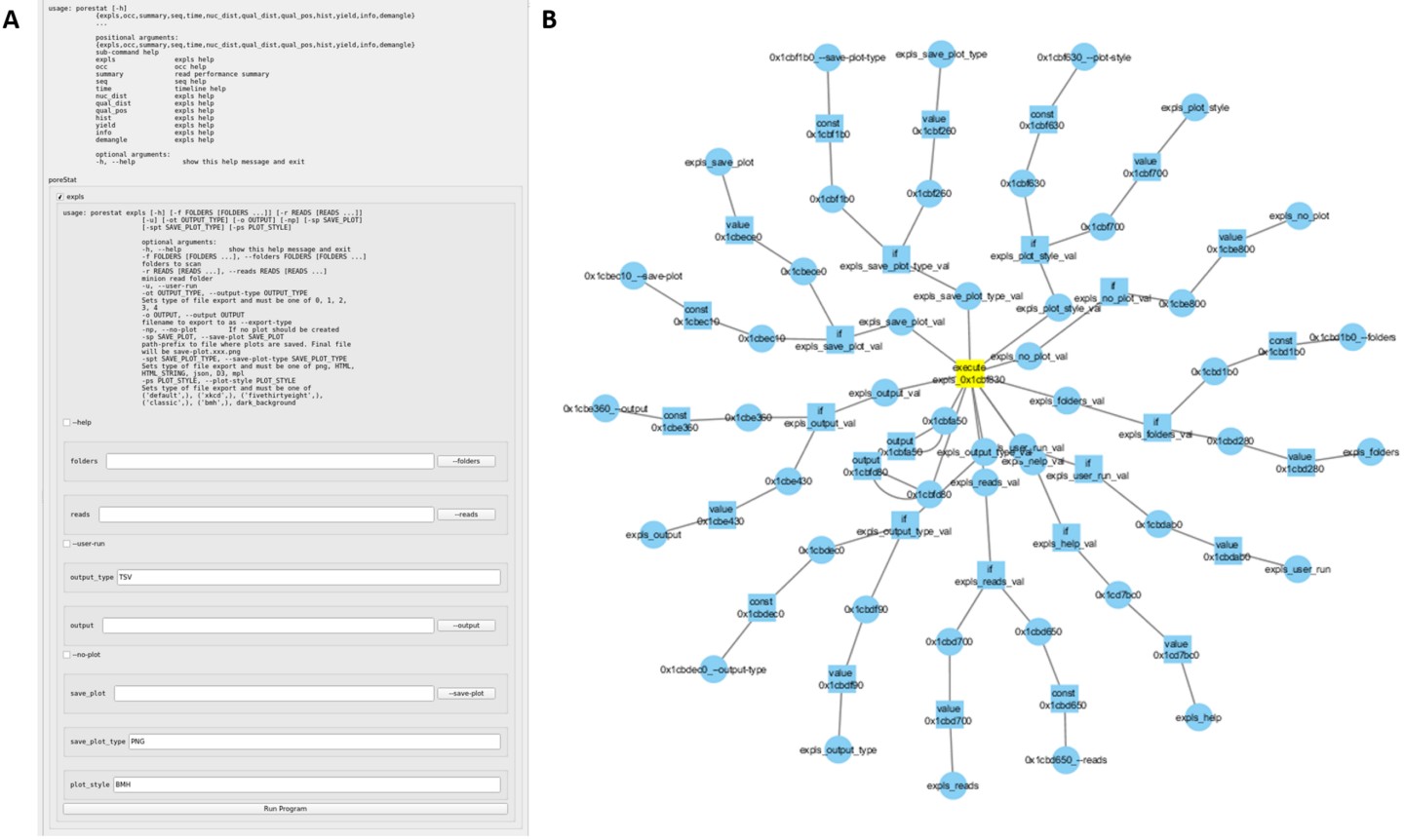

**Figure A3** (A) An automatically generated *bioGUI* template from the poreSTAT (Internal tool for minION sequencing analysis) python argument parser. (B) The resulting execution network for the *bioGUI* template shown in (A). The central node represents the fully assembled CL argument (yellow).                    

can be transferred to *bioGUI* via pipes. Of course, for both native and WSL processes, the output can also be transferred via netcat to *bioGUI*. The transfer of the GUI template within install modules is an example. If a process runs on a remote computer, the output can be transferred to *bioGUI* also via network, for example, netcat. Such a process can, for instance, be started by calling *ssh* from *bioGUI* with appropriate parameters. Finally *bioGUI* can also send HTTP POST requests to web-services and accepts an HTTP response as answer. This output can also be displayed by *bioGUI*.

Since the Docker engine is a local, native process, *bioGUI* also supports the use of Docker containers. The Circlator template is an example of how this can be implemented.

## Hardware specification for benchmarks

The relevant hardware for benchmarking *bioGUI* is summarized in Table A1.

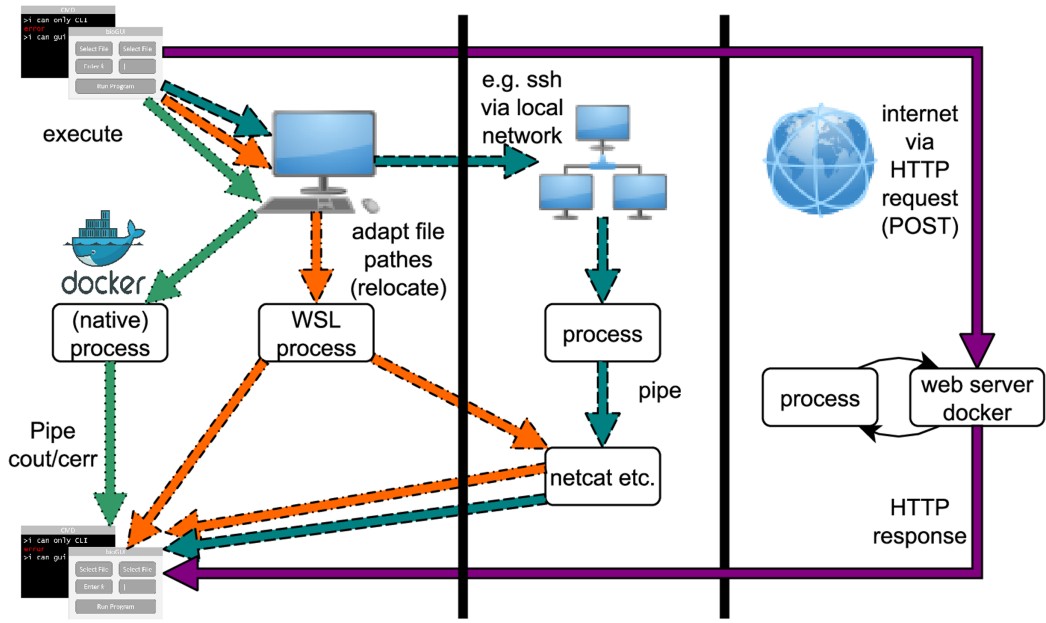

**Figure A4 Possibilities for running bioGUI: locally via processes, on a network via ssh or on the web via HTTP request/response.** Straight arrow (purple): HTTP execution mode; Dotted arrow (green): Docker execution; Dotdashed arrow(orange): bash/WSL execution; Dashed arrow(cyan): remote/ssh execution.

Table A1 Hardware used to benchmark *bioGUI*.

| Computer name | CPU | RAM | Storage |
|---|---|---|---|
| Linux server | Intel Xeon W-2145 CPU @ 3.70 GHz 8 cores (+8 HT cores) | 128 GB | Samsung SSD 860 1 TB SSD |
| Lenovo laptop (T470p) | Intel Core i7-7820HQ @ 2.9 GHz 4 cores (+4 HT cores) | 32 GB | Samsung MZVLB1T0HALR 1 TB SSD |
| Microsoft surface book | Intel Core i5-6300U @ 2.4 GHz 2 cores (+2 HT cores) | 8 GB | Samsung MZFLV128HCGR 128 GB SSD |
| Apple MacBook Air (mid 2012) | Intel Core i5 @ 1.7 GHz | 8 GB | 128 GB SSD |

## Template access

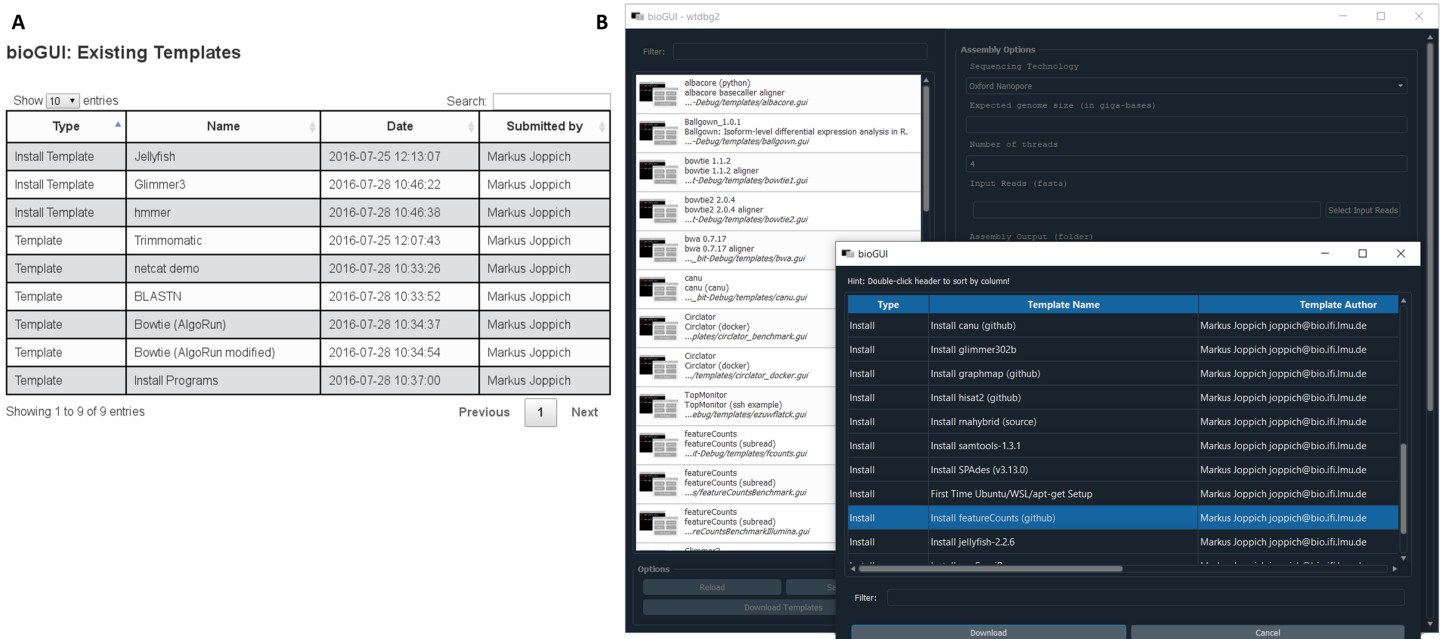

**Figure A5** (A) On our website a list of already existing templates can be browsed. Besides the description and author, also the type (install module or template) is shown. (B) All uploaded templates can be downloaded directly from within bioGUI. *bioGUI* allows to search in/filter all available install modules and templates.

## User survey

A user survey among 10 participants (four bioinformaticians, six collaborators (consisting of two under-graduate bioinformatics students and four external collaborators)) has been performed. The derived results are shown in Table A2 and the raw data are shown in Table A3.

**Table A2** Derived user survey results from the given answers (Table A3).

|  | *n* | Median | Mean | *p*-value | Variance |
|---|---|---|---|---|---|
| Better interface bio | 3 | 3 | 3 |  | 4 |
| Better interface collab | 6 | 4.5 | 4.5 |  | 0.3 |
| Better interface all | 9 | 4 | 4.00 |  | 1.75 |
| Easy to align CLI | 10 | 3.5 | 3.50 |  | 1.833 |
| Easy to align bioGUI | 10 | 5 | 4.90 | 0.0098 | 0.1 |
| Easy to install CLI | 10 | 5 | 4.40 |  | 0.711 |
| Easy to install bioGUI | 10 | 5 | 4.80 | 0.2023 | 0.178 |

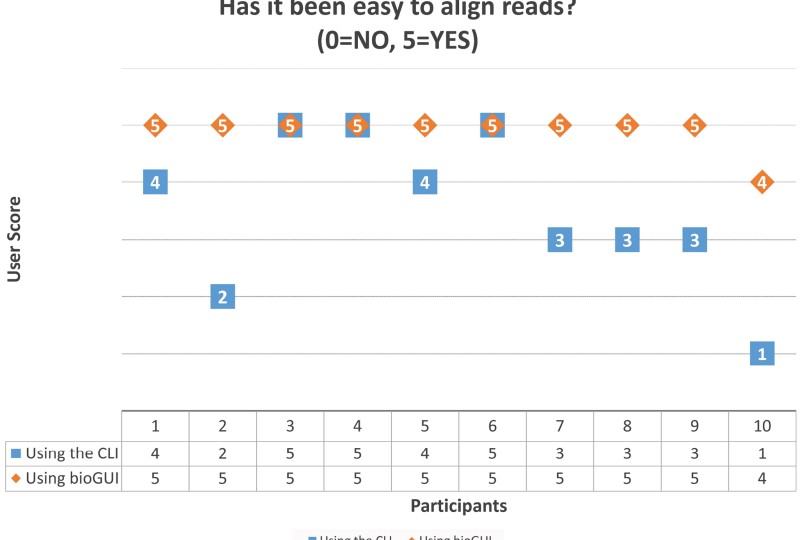

**Figure A6 Scores given by the 10 participants on the question "Has it been easy to align the reads?" after performing the task using the CLI and *bioGUI*.** These results show that most participants found the task easier using *bioGUI*, but for no-one it was harder to use *bioGUI*.

**Table A3 Relevant participant answers for the performed user survey and the results in Table A2.**

| | Participant 1 | Participant 2 | Participant 3 | Participant 4 | Participant 5 | Participant 6 | Participant 7 | Participant 8 | Participant 9 | Participant 10 |
|---|---|---|---|---|---|---|---|---|---|---|
| Usertype (0 = bioinformatician, 1 = student, 2 = collaborator) | 0 | 0 | 0 | 0 | 1 | 2 | 2 | 1 | 2 | 2 |
| Which kind of user-interface does the tool have? | CLI | CLI | CLI | CLI | CLI | CLI | GUI | CLI | CLI | GUI |
| What were the most struggle-some tasks in accessing and using the software? | Dependencies | Using software finding settings | Finding settings options needed | Installing software | Finding settings options needed | Dependencies starting the software using the software | Finding settings options needed | Finding settings options needed | Using the software | Finding settings options needed |
| CLI: Has the installation process been easy? (0 = NO, 5 = YES) | 4 | 3 | 5 | 5 | 5 | 5 | 4 | 5 | 5 | 3 |
| CLI: Has it been easy to align the reads? (0 = NO, 5 = YES) | 4 | 2 | 5 | 5 | 4 | 3 | 3 | 5 | 3 | 1 |
| bioGUI: Has the installation process been easy? (0 = NO, 5 = YES) | 5 | 5 | 5 | 4 | 5 | 5 | 5 | 5 | 5 | 4 |
| bioGUI: Has it been easy to align the reads? (0 = NO, 5 = YES) | 5 | 5 | 5 | 5 | 5 | 5 | 5 | 5 | 5 | 4 |
| Overall: Which interface was easier to use in your opinion? (0 = CLI, 5 = GUI) | 5 | 1 | 3 | 4 | 5 | 5 | 4 | 4 | 5 | |

## ACKNOWLEDGEMENTS

We thank Luisa F. Jimenez-Soto and Gergely Csaba for their valuable input as well as for reviewing the manuscript. We thank the participants in our survey for their time. We thank the reviewers for their constructive feedback.

### Funding

This work was supported by the Deutsche Forschungsgemeinschaft (Collaborative Research Centre SFB 1123-2/Z2). The funders had no role in study design, data collection and analysis, decision to publish, or preparation of the manuscript.

### Grant Disclosures

The following grant information was disclosed by the authors:
Deutsche Forschungsgemeinschaft (Collaborative Research Centre): SFB 1123-2/Z2.

### Competing Interests

The authors declare that they have no competing interests.

### Author Contributions

- Markus Joppich conceived and designed the experiments, performed the experiments, analyzed the data, prepared figures and/or tables, authored or reviewed drafts of the paper, approved the final draft.
- Ralf Zimmer conceived and designed the experiments, analyzed the data, prepared figures and/or tables, authored or reviewed drafts of the paper, approved the final draft.

### Data Availability

The bioGUI documentation is available at https://biogui.readthedocs.io/en/latest/. In order to setup Windows Subsystem for Linux (required for using bioGUI on Windows), follow the steps at https://biogui.readthedocs.io/en/latest/build_wsl.html. bioGUI is open source software. Releases and code are available at https://github.com/mjoppich/bioGUI. Additional software (cwl2biogui) is available at https://github.com/mjoppich/bioGUItools.

### Supplemental Information

Supplemental information for this article can be found online at http://dx.doi.org/10.7717/peerj.8111#supplemental-information.

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
