# Peer review of "From command-line bioinformatics to bioGUI"

_PeerJ, doi:10.7717/peerj.8111_

## Round 0.1 · original submission · Major Revisions

Dear Drs. Joppich and Zimmer

Thanks for submitting your manuscript to PeerJ. I have now received three independent reviews of your work, and as you will see, one reviewer (#1) recommended rejection, while the other two suggested major revisions. I am affording you the option of revising your manuscript according to all three reviews, but understand that your resubmission will likely be sent to at least one new reviewer for a fresh assessment.

The reviewers raised many concerns about the manuscript. There seems to be general agreement for a lack of hypothesis testing, as well as data and examples of bioGUI superiority in relation to other approaches. It is certainly disappointing that the reviewers could not implement some of bioGUI’s templates. If usability is a problem for these reviewers (all of which are experts in this field) it will obviously be a bigger issue for the general audience. Please fix all of these issues raised by the reviewers.

Please also eliminate unnecessary language that assumes bench scientists are not skilled at informatics (per reviewer3’s concerns). This is outdated as well as stereotypical.

In short, bioGUI does not seem like it went through rigorous testing by colleagues prior to submission. Please consider doing this before resubmission, and also providing demonstrations of usability and robust comparisons with other approaches.

I encourage you to revise your manuscript accordingly, taking into account all of the concerns raised by both reviewers, as well as myself.

Good luck with your revision,

-joe

Reviewer 1 ·

Basic reporting

This manuscript presents bioGUI for running bioinformatics software on Windows under the Windows Subsystem for Linux). Usability is a serious issue for software - particularly in the discipline of bioinformatics which includes a wide variety of software with even wider variability in long-term support. While the authors have identified an important need, it is difficult to understand how bioGUI would alleviate this problem. (1) The manuscript lacks a mature review of the current state of the field, listing only two workflows - one of which (Yabi) is sadly out of date. (2) The manuscript sets up CLI-aware vs. GUI-dependent end users as the two use cases; in reality, the distribution is much more nuanced as many "wet-lab" researchers are familiar enough with CLI to install software and run basic commands. No evidence is presented to support such a binary distinction. (3) There is no methodology described or referenced that explore the actual trouble spots in software usability except the authors stating that it is installation and use (which are vague enough to be generally useless). (4) There is no methodology for assessing whether bioGUI actually does make it easier to run bioinformatics software despite multiple statements that it does.

Experimental design

This manuscript describes a software package; as such, a traditional experimental design is not to be expected. However, from a usability standpoint the manuscript does not appear to design any experiments to determine appropriate trouble spots in bioinformatics software use. In addition, there are no experiments presented to assess the success of their software.

Validity of the findings

A stated in previous sections, there is no data presented here that can be evaluated. On first principles, it appears that bioGUI relies heavily on the implementation of Windows Subsystem for Linux that is developed by Microsoft. However, this is incidental to the value of bioGUI, since any bioinformatics software can take advantage of the WSL. bioGUI appears to replace some CLI (software install and command run) with other command-like language (i.e., XML specifications) but it is not clear how (or if) this would improve usability. Finally, usability metrics would have been useful for gauging the actual value of bioGUI, but nothing is discussed in this regard.

·

Basic reporting

The authors created a tool that allow users to generate graphical interfaces for existing bioinformatics tools, currently, only available through command line.
The article is well written and supply a good explanation regarding this tool importance. It is a relevant work and I am certain that it will help the community to gain access to bioinformatics, as much as, will help the developers to spread their tools.
In the current version, however, the tools seems limited to windows users, depending heavily on the integration through WSL (Linux subsystems). Moreover, after trying the tool, I got a few concerns regarding the range of its capabilities. I believe the paper could be more focused on which tools are currently supported and how they are integrated to the GUI and less on the importance of Graphical User Interface.
Below, I will point out the aspects that I believe need to be addressed in order for this work to be published.

Experimental design

No comments.

Validity of the findings

No comments.

Additional comments

In the introduction, the authors describe the problems of using tools like AlgoRun and among them they state that docker solutions to run this software depends on sophisticated configurations, rather than the simple WSL download. I disagree with such statement, as the simplicity to install bioinformatic tools on WSL will also depend on the characteristics of the WSL version and the installation of dependencies for the tool of interest. Such problems have been mentioned in the paper Morais et al., 2018 "BTW—Bioinformatics Through Windows: an easy-to-install package to analyze marker gene data." PeerJ 6 (2018): e5299, where some tools wouldn't work at all in the WSL, because of compilation issues. The authors of the aforementioned paper use the same strategy of bash scripts to install bioinformatic tools in WSL and there is already reports of complications depending on the tools being installed.

Although the authors mention the multi-platform nature of bioGUI (in the lines 62 and 121), there is way more emphasis on the windows platform in the abstract and in the online documentation for usability. Giving the impression that although it was designed in a cross-platform language, there is only support for Windows users. Is it possible to download it and run it from Mac and Ubuntu (or other Linux distribution)?

Regarding the XML templates for GUI, would be nice to have a figure or scheme with a simple example like trimmomatic and how a XML template should look like.
One suggestion to increase reproducibility, would be to have a log file of the bioinformatic commands used or some button (template example) for outputting this log file with the selected parameters for each bioinformatic tool.

After installing the tool in a Windows 10 machine, Build 15063, and running the program bioGUI, I found some templates, but was not able to use the available tools. Checking the user's guide, I found that it supplies initial instructions until the download of templates and installation of software, but it doesn't explain how to use the newly installed software. Also, when checking the folders under WSL to see if the new software was there, nothing was found. Apparently the installation had some issue that was not even reported.

The user's guide (https://biogui.readthedocs.io/en/latest/user_guide.html) also mentions a list of available templates but when we click on the link (https://www.bio.ifi.lmu.de/software/bioGUI), we are sent to an error page.

Moreover, when checking the list of available software, there is currently 15 tools available through bioGUI (none of which were actually running after my installation), this is almost the same number offered by SEED 2 by (Vetrovsky et al., 2018) SEED 2: a user-friendly platform for amplicon high-throughput sequencing data analyses. Bioinformatics. 2018 Feb 14;34(13):2292-4. The mentioned tool is focused on amplicon data analysis, which is a limited sub-field of bioinformatics. If bioGUI is meant to be a tool for the whole spectrum of tools in bioinformatics and if it is really as easy as a few minutes of work for someone to create a template (lines 64 and 65), I believe there should be way more templates and tools available to the community.

Small remarks:
I feel that for the broad audience and knowing that bioinformatic software is also made by domain-experts, would be interesting to include some brief explanations of the terms "Continuous Integration" and some examples of use, as a simple template creation example.

line 206 "reproducable" > "reproducible".

Reviewer 3 ·

Basic reporting

No Comment.

Experimental design

No comment.

Validity of the findings

Joppich and Zimmer have identified the abundance of command line applications for bioinformatics as a common issue for life scientists who are not experts in the use of unix based computer systems. The authors of this manuscript aim to lower the barrier to entry for scientists by providing a program (bioGUI) to easily convert command line applications to a point-and-click format. While a user test case and a developer test case are described in the paper and appendix, data-driven assessment of bioGUI in the context of life scientist laboratory use and developer use are not described in the manuscript.

A theme within this manuscript is the use of laptops or laboratory computers for common bioinformatic analyses, such as high throughput sequencing analysis. The researchers use Galaxy and Yabi as examples of existing workflow systems for common laboratory analyses. Galaxy is often installed on compute nodes more powerful than the standard laboratory computer so as to handle the large size of the datasets generated by standard sequencing platforms and the compute power required to analyze these datasets appropriately. Have the researchers tested a “typical” laboratory computer for its use in the test cases described in the manuscript? The use case should include run time statistics including required storage, RAM and time required for the analysis to be completed. These requirements should be compared to a “typical” laboratory computer. Could access to a remote server or compute cluster be integrated into the templates by developers?

Additionally, while basecalling during MinIon data acquisition is one use special use case, describing the use of bioGUI with more common use cases (such as Illumina-based genome assembly, or RNA-seq analysis) would greatly strengthen the paper.

Additional comments

This manuscript contains combative language toward “wetlab” life scientists. It should be noted that life scientists are not incapable of using unix-systems and command-line based programs but rather their expertise is concentrated in other disciplines. Please address the language in the manuscript for such bias.

---

## Round 0.2 · Minor Revisions

Dear Drs. Joppich and Zimmer

Thanks for revising your manuscript. Reviewer 1 has provided more suggestions, and overall it appears that improvements to English and grammar are still needed.

I encourage you to revise your manuscript, accordingly, taking into account all of the new concerns raised by the reviewer, and to consider having an English expert read over your revision. I do think your work will be ready for acceptance once these minor issues are addressed.

Good luck with your revision,

-joe

Reviewer 1 ·

Basic reporting

1. Basic Reporting
This manuscript is improved from the original submission by the addition of a benchmarking study and numerous text changes. It represents a fairly comprehensive description of the authors' software, bioGUI. Some clarity has been added surrounding the different target audiences - developers and end-users - and this is helpful. A few concerns remain. (1) There is no methodology (described or referenced) that explores the actual trouble spots in bioinformatics software usability. This could be done with a fairly straightforward usability study or user survey. The lack of data to support the authors' fundamental assumptions detracts from the impact of the manuscript. The authors have added two citations (line 175), but these refer to personal communications that are not possible to evaluate. (2) The authors have added several benchmarking results that demonstrate well the function of their software on various hardware/OS combinations; however, there is no methodology for assessing whether bioGUI actually makes it easier to run bioinformatics software (e.g., pre- and post-testing). (3) The writing style is awkward, with dropped definite articles, misspelled words, and difficult phrasing throughout.

Specific comments:

Line 11 and beyond: An abbreviation for "command-line" (CL) is defined in line 10 and can be used throughout the rest of the manuscript.
Line 43: skeptical is misspelled.
Line 49: "when not yet running a software routinely on many data yet" - this phrasing is awkward and difficult to understand.
Line 63: GUI is already defined on line 41.
Line 95: "highly sensible" does not make sense in this context. is this supposed to be "highly sensitive" instead?
Line 109: CWL is already defined.
Lines 127-128: the use of italics here is not clear; if this is meant to be a direct quote, it should be surrounded by quotation marks.
Line 136: please replace "his" with "their."
Line 177: "command-line line" is redundant.
Lines 206-207: extra "the" between installed and software; also the last part about already being derived is confusing: "the path to the installed the software as well as other default values that can already be derived during the installation"
Lines 225-226: the reference to a petri net is not clear and should be removed, unless the authors propose that bioGUI can be treated mathematically as a bipartite directed graph to some advantage.
Line 246: execution is misspelled.
Line 338, 352-354: please clarify the interchangeable use of brew and homebrew. the package manager is called homebrew, while the executable for that manager is called brew.
Line 347: "from on" should be either "from" or "on" but not both.
Line 362: sub-processes should be singular.
Line 504: existent is misspelled.
Line 505: please change no to not.
Line 516: please change themselves to herself.
Line 527: analysis and hence are misspelled.
Line 544-546: the last two sentences of this paragraph are quite confusing and contain many hyphenated words that do not appear to require hyphenation.
Line 571: please replace "which's output" with "and then the output".

Experimental design

No comment

Validity of the findings

No comment

·

Basic reporting

No comments.

Experimental design

No comments.

Validity of the findings

No comments.

Additional comments

I am glad to see that all my concerns were addressed and the paper was extended considerably. I am also happy to see that the general supporting material for the software was expanded. I am sure bioGUI is a very useful tool for the whole scientific community.

---

## Round 0.3 · accepted · Accept

Dear Drs. Joppich and Zimmer:

Thanks for re-submitting your revised manuscript to PeerJ, and for addressing the concerns raised by the reviewers. I now believe that your manuscript is suitable for publication. Congratulations! I look forward to seeing this work in print, and I anticipate it being an important resource for non-computational scientists conducting bioinformatics analyses.

Thanks again for choosing PeerJ to publish such important work.

-joe

Reviewer 1 ·

Basic reporting

No comment

Experimental design

The Methods section begins with several paragraphs describing systems already in existence. This information might be better placed in the Introduction as it does not describe methodology developed in this manuscript.

Validity of the findings

No comment.